# The Role of Biophysical Factors in Organ Development: Insights from Current Organoid Models

**DOI:** 10.3390/bioengineering11060619

**Published:** 2024-06-18

**Authors:** Yofiel Wyle, Nathan Lu, Jason Hepfer, Rahul Sayal, Taylor Martinez, Aijun Wang

**Affiliations:** 1Department of Surgery, School of Medicine, University of California-Davis, Sacramento, CA 95817, USA; ybwyle@ucdavis.edu (Y.W.); nhlu@ucdavis.edu (N.L.); jhepfer@ucdavis.edu (J.H.); rsayal@ucdavis.edu (R.S.); tgmartinez@ucdavis.edu (T.M.); 2Institute for Pediatric Regenerative Medicine, Shriners Children’s, Sacramento, CA 95817, USA; 3Department of Biomedical Engineering, University of California-Davis, Davis, CA 95616, USA; 4Center for Surgical Bioengineering, Department of Surgery, School of Medicine, University of California, Davis, 4625 2nd Ave., Research II, Suite 3005, Sacramento, CA 95817, USA

**Keywords:** organoid, bioengineering, mechanobiology

## Abstract

Biophysical factors play a fundamental role in human embryonic development. Traditional in vitro models of organogenesis focused on the biochemical environment and did not consider the effects of mechanical forces on developing tissue. While most human tissue has a Young’s modulus in the low kilopascal range, the standard cell culture substrate, plasma-treated polystyrene, has a Young’s modulus of 3 gigapascals, making it 10,000–100,000 times stiffer than native tissues. Modern in vitro approaches attempt to recapitulate the biophysical niche of native organs and have yielded more clinically relevant models of human tissues. Since Clevers’ conception of intestinal organoids in 2009, the field has expanded rapidly, generating stem-cell derived structures, which are transcriptionally similar to fetal tissues, for nearly every organ system in the human body. For this reason, we conjecture that organoids will make their first clinical impact in fetal regenerative medicine as the structures generated ex vivo will better match native fetal tissues. Moreover, autologously sourced transplanted tissues would be able to grow with the developing embryo in a dynamic, fetal environment. As organoid technologies evolve, the resultant tissues will approach the structure and function of adult human organs and may help bridge the gap between preclinical drug candidates and clinically approved therapeutics. In this review, we discuss roles of tissue stiffness, viscoelasticity, and shear forces in organ formation and disease development, suggesting that these physical parameters should be further integrated into organoid models to improve their physiological relevance and therapeutic applicability. It also points to the mechanotransductive Hippo-YAP/TAZ signaling pathway as a key player in the interplay between extracellular matrix stiffness, cellular mechanics, and biochemical pathways. We conclude by highlighting how frontiers in physics can be applied to biology, for example, how quantum entanglement may be applied to better predict spontaneous DNA mutations. In the future, contemporary physical theories may be leveraged to better understand seemingly stochastic events during organogenesis.

## 1. Introduction

As the human embryo develops, mechanical forces exerted by the uterus, amniotic fluid, and surrounding tissue physically shape each developing organ system. Dysregulation of these forces can lead to severe and often fatal consequences. A striking example is congenital diaphragmatic hernia (CDH), where incomplete closure of the diaphragm allows abdominal viscera, such as the liver and intestines, to herniate into the thoracic cavity and physically compress the developing lung [1]. CDH kills 37% of affected babies and leaves survivors with lifelong respiratory complications such as chronic lung disease and pulmonary hypertension [1,2]. Current therapies for CDH involve biophysical intervention: inserting a balloon into the trachea to increase the intrapulmonary pressure. This increased pressure exerts a stretching force on the fetal lung, which activates lung cell growth [3,4]. Despite the demonstrated importance of mechanical biology in human health, physiological research has traditionally concentrated on biochemical pathways in organ development, maintenance, and disease. However, advances in microscopy and computational modeling are allowing for a more robust characterization of the biophysical forces underlying organ development and homeostasis [5]. Concurrently, improvements in materials science and bioengineering have generated stem cell-derived models of organ development, or organoids, which can incorporate these forces to more faithfully model in vivo organogenesis [6]. In this review, we highlight how engineering principles have been applied in tissue culture to investigate biophysical parameters, including size, shape, and stretch [6]. Incorporating these forces into organoid technologies will generate more translatable and potentially transplantable tissue using patient-derived stem cells. Current organoid systems are limited to fetal-like states; therefore, transplanted organoid therapies may make their first clinical impact in the realm of fetal medicine. While this review aims to highlight recent advancements in biophysics related to organogenesis, it does not cover some key frontiers in biophysics including biophysical regulation of the immune system [5,7] piezoelectric biomaterials [8], or the role of electrical stimulation in stem cell culture [9], which have been recently reviewed in the provided references.

## 2. Clinical Demand for Improved In Vitro Organoid Models

In total, 90% of clinical candidate drugs fail to reach the market at the transition from preclinical to clinical applications, colloquially termed the “Valley of Death” [10]. These high attrition rates underscore a disconnect between preclinical models and actual patient responses to treatments [11,12]. Compared to traditional 2D cell culture or rodent models, organoids better recapitulate human physiology, providing a more reliable and higher throughput model to screen the safety and efficacy of new drugs [13,14]. Traditional organoid systems utilized “differentiation media”, consisting of growth factors and small molecules, to chemically induce stem cells into differentiated tissues expressing similar markers to native organs [15]. By recreating the biochemical niche each organ experiences during embryonic development, biologists have developed chemical cocktails that can differentiate stem cells into tissue resembling nearly every organ system, including the brain [16], heart [17], lung [18], liver [19], kidney [20], and pancreas [21]. These biochemical approaches, however, largely ignored biophysical factors underlying organ development, resulting in heterogeneous and unreproducible culture systems [6]. Recent advances in material sciences and bioengineering have enabled culturing systems that can better approximate the biophysical niche experienced during organogenesis, yielding more physiologically relevant tissues [22,23].

## 3. Principles of Biophysical Factors in Regulating Organ Development

### 3.1. Stiffness, Viscoelasticity, and Shear Forces in Organ Development, Homeostasis, and Pathology

Cells and the extracellular matrix that they inhabit are defined as viscoelastic materials, meaning they exhibit both time-dependent viscous and instantaneous elastic behavior when subjected to forces [24]. When a constant force is applied to purely viscous material, it will resist movement and dissipate heat energy over time. Perfectly elastic materials, however, immediately store energy during deformation [25]. A dynamic balance between viscous and elastic forces plays a critical role in organogenesis. For example, viscoelasticity mediates the formation of the intestinal crypt [26], and elastic forces facilitate neuronal plasticity and outgrowth in the developing hippocampus [27]. In lung organoids, increased viscoelasticity promotes airway-mimetic tubular morphology [28] and may impact alveolar growth [29]. Increasing viscoelasticity in the microenvironment has been shown to increase proliferation and morphogenesis in ex vivo models of the kidney [30], liver [31], cartilage [32], and brain [33].

While viscoelasticity describes a material’s ability to deform and recover under an applied force overtime [34,35], stiffness—or Young’s Modulus—is an instantaneous description of a material’s ability to resist deformation after applying a force [36,37] (Figure 1). Physiological stiffness is a defining characteristic in pathological states, such as cancer, where stiffer microenvironments induce tumorigenic and metastatic phenotypes [38,39]. Cancer cells actively remodel the extracellular matrix by increasing the cross-linking density and altering the composition of matrix proteins, such as collagen [40,41], creating a positive feedback loop of cancer-mediated stiffness and stiffness-mediated cancer. Persistent changes in microenvironmental stiffness may explain high rates of cancer recurrence after tumor removal or chemotherapy [42]. The impact of matrix viscoelasticity and stiffness on stem cell differentiation and growth has been recognized for nearly two decades [43]; however, bioengineers have only recently developed materials that can independently tune viscoelasticity without impacting stiffness, and vice versa [44]. Under constant stiffness, increasing viscoelasticity promotes cell spreading, proliferation, and osteogenic differentiation of mesenchymal stem cells [45]. Conversely, decreasing stiffness increases chondrocyte proliferation when viscoelasticity is held constant [46].

Beyond stiffness and viscoelasticity, shear forces play a formative role in organ development, homeostasis, and disease—best exemplified by renal physiology. While viscoelasticity does appear to regulate nephrogenesis [61], shear forces play a fundamental role in kidney development [62]. Serluca et al. demonstrated that vascular flow is required for glomerular assembly in zebrafish, and replacing blood with saline did not significantly impact glomerulogenesis, indicating that shear forces were necessary and sufficient for kidney morphogenesis [62]. Under homeostatic in vivo conditions, the kidney interprets changes in shear forces to modulate blood volume and pressure by initiating hormonal cascades via the renin-angiotensin-aldosterone system [63]. In organoid models of pathology, such as polycystic kidney disease, fluid shear stress drives cyst development by promoting increased glucose absorption [64]. In the developing brain, shear forces drive motor neuron differentiation and function [65]. If the structural integrity of the blood–brain-barrier is compromised, disruptions in neuronal shear stress, tension, and compression drive the pathogenesis of Alzheimer’s disease (AD) [66]. These clinical observations have been recapitulated using organoid models of AD [67]. Moreover, fluid flow is necessary to sustain the mitotic activity of gastric epithelial cells [68]. Shear forces thus impact development, normal physiology, and can contribute to disease when dysregulated.

### 3.2. Mechanotransductive Signaling Pathways: YAP/TAZ and the Intersection between Biophysical and Biochemical Pathways

The Hippo-Yes-associated protein/Transcriptional co-activator (Hippo-YAP/TAZ) signaling pathway is a key mechanotransductive pathway that governs tissue formation and maintenance. Matrix stiffness, sensed through integrin-mediated focal adhesion assembly and actin cytoskeleton tension, leads to dephosphorylation and nuclear translocation of YAP/TAZ [69]. Inside the nucleus, YAP/TAZ interacts with TEAD transcription factors [70] and influences master regulators including Wnt, Notch, and TGF-β [71,72] (Figure 2A). Disruptions in the biophysical niche, such as aberrant stiffness, can cause persistent activation of the Hippo pathway, constitutive activation of growth pathways, and cancer phenotypes [73]. YAP activation downstream of matrix stiffening has been shown to induce VEGF secretion, promoting angiogenesis, and likely contributes to metastasis via the tumor microenvironment [74,75]. In the context of embryogenesis, YAP plays a fundamental role in establishing developmental axes and is necessary for cell migration during gastrulation by regulating cytoskeletal organization in response to intracellular tension [76]. YAP/TAZ has also been shown to drive organ patterning and development downstream of geometric constraint via the YAP-Notch axis [77,78] and resist gravitational forces during organogenesis [79]. Beyond tension sensing, YAP responds to changes in flow shear stress, mediating a cellular response towards osteogenesis in mesenchymal stem cells cultured in a high-flow environment [80].

YAP is a relatively well-characterized, mechanically transductive signaling pathway but represents only one of many signaling cascades activated by biophysical forces. In response to embryonic compression, for example, Twist apically constricts ventral cells and mediates mesoderm invagination [81]. During organogenesis, matrix remodeling activates β1 integrin, vinculin, and actin polymerization to promote cellular adhesion and proliferation [82]. Later in development, the cyclic stretch in the developing lung triggers a cascade of molecular events, including the activation of histone deacetylases, miRNA, long-noncoding RNA, and critical factors such as TGFβ, αSMA, and PDGFRA [82]. The cyclic stretch also mediates the release of serotonin from neuroendocrine cells [83] and induces surfactant production and maturation in type II alveolar pneumocytes [84,85]. Other mechanotransducers essential for organ development and maintenance include integrins [86], piezo channels [87], transient receptor potential channels [88], cadherin [89], plectin [90], talin [91], vinculin [92], filamin [93], and dystrophin [94].

**Figure 2 bioengineering-11-00619-f002:**
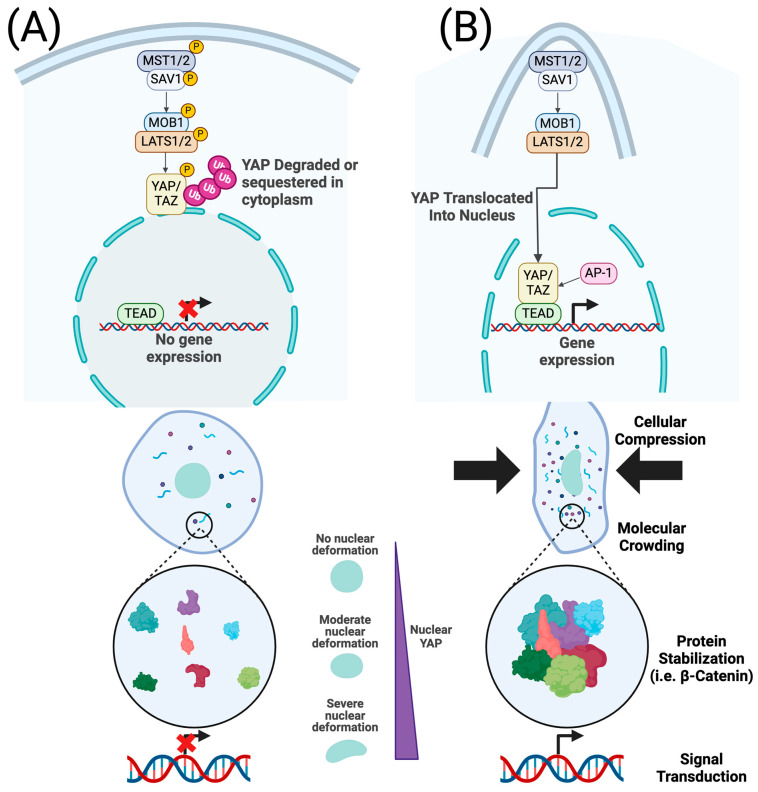
Mechanotransduction mediated by YAP/TAZ signaling and molecular crowding. (**A**) In the absence of cellular compression, which may occur in softer microenvironments, YAP/TAZ is phosphorylated and degraded [69]. Cytoplasmic molecules are relatively dispersed, and the nucleus is not deformed. (**B**) Cellular compression, which can be mediated by stiffer microenvironments, inhibits the activity of HIPPO signaling pathway kinases, allowing YAP/TAZ to translocate into the nucleus, bind to TEAD, and activate downstream target genes (i.e., Wnt, Notch, and TGF-β) [69,70,71]. Nuclear deformation further increases YAP nuclear translocation, in part by increasing the nuclear pore size. Cellular deformation results in molecular crowding, stabilization and accumulation of protein complexes, and activation of downstream targets [95,96].

In addition to mechanically sensitive genetic pathways, physical alterations in cell morphology can inherently influence signaling cascades. Liquid–liquid phase separation (LLPS) is a physical phenomenon where molecules, originally in a homogenous mixture, will segregate into distinct liquid compartments [95]. In the context of cellular biology, LLPS can create membraneless organelles, such as nucleoli, stress granules, and P bodies, which play crucial roles in development and tissue homeostasis by regulating protein synthesis, RNA metabolism, and the cellular stress response [95]. When cells are compressed, local molecular concentrations will increase as cytoplasmic volumes decrease in a process called molecular crowding. Molecular crowding can potentiate phase separation by altering the local biochemical and biophysical niche within cytoplasmic domains [95]. Moreover, increasing the local protein concentration can stabilize protein complexes, increase receptor-ligand kinetics, and drive signaling events [96] (Figure 2B). Increased local concentrations of β-catenin downstream of compression enable nuclear translocation in crowded domains [95,96]. Nuclear β-catenin activates the Wnt signaling cascade, leading to cell survival, proliferation, and differentiation, independent of genetic cascades mediated by YAP mechanotransduction [96]. Molecular crowding has also been implicated in chemoresistant pancreatic cancer [97]. Finally, mechanical forces can physically deform the nucleus, which independently serves as an axis of control for cell signaling and function [98].

### 3.3. Geometric Constraints: Size and Shape

Engineered approaches to cell culture have revealed that size constraints within planar and volumetric spaces are critical in shaping stem cell development. In the recent decade, surface micropatterning of proteins onto glass and plastic substrates has enabled precise control over organoid size, number, and shape, mitigating some of the stochasticity of earlier organoid protocols, which relied solely on self-organization [99,100]. Photolithography can now generate 2D patterns below single-micron resolution, permitting the investigation of size and shape within the context of tissue growth and identity [101]. Early 2D patterning work revealed that confining stem cells into disk shapes generated spatial patterns of gene expression, reminiscent of germ layer patterning [102]. Ribiero et al. showed that increasing the length/width aspect ratio of individual cardiomyocytes increased contractility by increasing sarcomere alignment [103]. The size of tissue patterns has also been shown to be critically important in determining cell identity. For example, differentiating pluripotent stem cells on 250 µm micropatterns generated tissues with polarized neural rosettes, whereas cells seeded on 150 µm diameter patterns generated spinal neuroepithelial tissue in the same biochemical environment [104]. Karzbrun and colleagues showed that controlling the size and shape of stem cells during neural induction can mediate spontaneous folding into neural tube-like structures [105]. Many of these organoid systems employed a “2.5D” cell culture approach, in which cells are first seeded onto protein patterns to form a 2D monolayer before a matrix material, such as Matrigel, is added on top of the monolayer, facilitating a transition into 3D structures [99,105].

During in vivo organogenesis, differential growth between developing layers of tissue generates mechanical stress, physically shaping nascent organs in all three dimensions [106]. Mesenchymal condensation around the developing epithelium generates mechanical stability or instability and influences tissue morphology and gene expression [107,108]. Branching morphogenesis in the developing lung, for example, is mediated by physical patterning between the budding epithelium and surrounding tissues [109]. Recent advances in bioengineering have facilitated the creation of materials with controlled microscale volumes in all three dimensions, with sufficient rigidity and porosity for bioreactor culture [110,111]. Ding and others generated a hyaluronic-based microcarrier system which emulates the biophysical niche of chondrocyte cells and found that 3D geometric constraints maintained chondrocyte stemness [112]. Other groups have used curved microfluidic channels to show that 3D shapes determine cell velocity, morphology, and collective migration behavior [113]. The geometry of the cellular microenvironment influences tensile stress of the cytoskeleton, which is mechanically transduced via actomyosin contractility [108], to physically change the shape of the nucleus [114]. Nuclear deformation can increase nuclear pore size and activate YAP/TAZ signaling cascades.

## 4. Approaches to Use Biophysical Factors in Regulating Organoid Development

### 4.1. 3D Printing Approaches to Tissue Patterning and Modeling

Bioengineers have leveraged advancements in 3D printing and biomaterials to generate 3D tissue structures which yield tissues with a more reproducible size and shape compared to unpatterned organoids [115]. These approaches offer high-throughput systems which control cellular patterning in three dimensions with bioinks that can be tailored to address a specific design constraint (Figure 3). Below, we briefly describe six of the most common bioprinting techniques and summarize the advantages and limitations of each approach.

(i) Stereolithography (SLA), invented by Chuck Hall in 1984, was the first 3D printing method developed and uses lasers to solidify photoreactive polymers one layer at a time [119]. With advances in material sciences in the recent 40 years, SLA can currently achieve an XY resolution of 50 µm on acrylate- or epoxide-based substrates [119]. Grigoryan and colleagues generated a vascularized alveolar model, complete with circulating red blood cells, using SLA-printed hydrogels [120]. While SLA is a well-established printing technique, relatively long UV exposure requirements and an inability to print with multiple materials limit printing with live tissues [119].

(ii) Selective Laser Sintering (SLS) is a 3D printing technique that uses a laser to fuse powdered material into a solid structure, layer by layer. In bioprinting, SLS has been utilized to generate dendritic vascular networks in cell-laden hydrogels [121] and biodegradable microspheres for lung organoids [122].

(iii) Material extrusion (ME) printing refers to fluid deposition modeling (FDM) and 3D dispensing [123]. FDM traditionally involves heating and extruding thermoplastics through a nozzle to generate plastic features [124]. FDM has been used to create patient-specific airway models to study airflow and aerosol deposition in human lungs [125,126,127]. Three-dimensional dispensing enables concurrent deposition of multiple cell types, enabling more complex tissue models. For example, Roth et al. created spatially controlled assembloids by dispensing neural organoids in close proximity to patient-derived glioma organoids [118].

(iv) Digital Light Processing (DLP) uses light to cure photo-sensitive resin layer by layer. New methacrylated resin materials allow for rapid, high-resolution printing: down to 22 µm [128,129]. DLP has been used to differentiate bone marrow stem cells for bone regeneration using rapidly printed microspheres [130]. Carberry and others used DLP to print thioether elastomers to create 3D arrays of intestinal stem cells [128].

(v) Laser-Assisted Bioprinting (LAB) is a technique that utilizes laser energy to induce the rapid transfer of material from a donor ribbon to a receiving substrate, allowing for high-resolution patterning of living cells and biomaterials with minimal damage to the cell [131]. LAB has also been used to print human umbilical veins and mesenchymal stem cell patches for cardiac generation [132].

(vi) Multi-Photon Polymerization (MPP) is currently the most precise 3D printing technique capable of achieving less than 100 nm resolution in droplet printing [116] and 5 μm resolution in biocompatible collagen gel. It is used to fabricate human capillary models that are true to scale [117]. MPP has also been used to generate microfluidic devices for dynamic cell culture which leverage the precise resolution of MPP to generate channels that achieve sub-microliter volume flow rates [133].

In summary, SLA offers the capability to fabricate organ models with remarkable detail, yet its dependency on ultraviolet (UV) light raises concerns regarding cell viability and functionality [119]. SLS enables the creation of intricate organ geometries without the necessity for support structures, enabling the construction of organ models with integrated functionalities [121]. However, the SLS process may lead to rough surface textures and requires meticulous material handling, potentially impacting biocompatibility [122]. ME provides versatility in the use of diverse materials, which is crucial for simulating the varied biophysical properties of organs [118]. Nonetheless, its comparatively slow speed and reduced resolution may hinder the precise replication of complex tissue microarchitectures [118]. DLP is distinguished by its rapid printing capabilities and high resolution, which are advantageous for preserving cell viability and enables the creation of high-throughput organoid systems [130]. However, the reliance on UV light in DLP could adversely affect living tissues, potentially altering cellular behavior and functionality [128]. LAB allows for precise cell placement with minimal damage, essential for tissue construct assembly, though its limited throughput and size limitations of printed structures may restrict its use in larger organ development projects [131]. MPP achieves unmatched precision in fabricating capillary networks and microfluidic devices vital for organ functionality [133], yet its scalability challenges, limited material compatibility, and slow fabrication speed pose significant barriers to its broader adoption [116,117].

### 4.2. Microfluidic Systems to Pattern Tissue Formation

Microfluidic chips are devices with microchannels that enable precise fluid manipulation, providing an ultra-high throughput and adaptable platform for cell culture. The IFlowPlate, for example, can culture 128 independently perfused vascularized organoids without any external pump [134], while OrganoPlate enables the culture of multiple microfluidic chips simultaneously, allowing multiplexed, ultra-high throughput screening [135]. Modern microfluidic chips can be manufactured for under 2 USD per device [136] and can support the culture of multiple cell types within multiple morphogen gradients [137]. Microfluidic systems have been employed to study blood–brain barrier dysfunction in the context of Alzheimer’s disease progression [67,138] and to investigate mechanical forces which underlie embryology [81]. Recently, Rousset et al. engineered a hanging drop micro/mesofluidic device designed to mimic the interaction between circulating cells and static organs, permitting detailed investigation into how recirculating tumor or immune cells interact with various organ systems [139]. Microfluidic platforms are now widely employed to investigate the systemic effects of diseases which impact multiple organ systems. These systems have provided critical insight into the pathology and treatment responses to infectious disease such as COVID-19 [23,140].

### 4.3. Assembloids and Vascularized Organoids to Study Biophysical Components’ Organ Assembly and Angiogenesis

An active field of research involves the fusion of multiple organoids to recapitulate subdomains within a single organ or crosstalk between multiple organ systems [141]. For example, the human heart, liver, bone, and skin tissue have been connected via an artificial recirculating vasculature system [142], providing a high-throughput, body-on-a-chip screening platform [143]. Assembloids are being leveraged to better understand fusion events which occur during embryonic development such as the assembly of the human cortex, hippocampus, and thalamus in human brain organoids [144]. Gabriel and colleagues generated a brain organoid that could detect light by fusing optic vesicles with a separately differentiated brain organoid [145], permitting an investigation of the structural organization and interaction between subdomains of the developing brain. More recently, Rawlings et al. generated endometrial assembloids which faithfully recapitulate key physical interactions between the embryo and endometrium [146]. The integration of multiple organoid systems permits exploration into biophysical tissue interactions that occur between tissues during organogenesis.

The development of vascular networks, or angiogenesis, is influenced by biophysical stimuli. For example, blood vessel branching and diameter are influenced by flow-induced shear forces [147,148]. Angiogenesis is also influenced by other biophysical parameters including stiffness [148], cyclic stretch [149], and 3D curvature of the microenvironment [150]. Specific mechanisms underlying this mechanotransducive are described in Section 3.2. Another key biophysical concept in organ development and maintenance is the diffusion of molecules, including oxygen, into the center of larger tissues. As organoid systems begin to scale up in size and complexity, there is an increased demand for systems that can adequately supply nutrients throughout larger tissues. Spinner or oscillatory bioreactors improve nutrient diffusion to an extent [151]; however, larger organoids still develop hypoxic cores as oxygen and nutrients are unable to penetrate into the center of dense tissues through passive diffusion [151]. Therefore, vascularized organoids are necessary to sustain even fetal organ-scale tissues [152,153]. Current vascularization strategies involve either (i) endothelial cell self-assembly or (ii) assembling pre-formed microvessels using materials lined with endothelial cells [154]. Vascularized brain organoids have been made by transplanting organoids into highly vascularized tissue in immunodeficient mice [155] or employing microfluidic devices which enable concurrent neural differentiation and vascularization, emulating in utero brain development [156]. Transplanting organoids into host animals supports the growth and development of larger organoid systems [155,157,158,159] and could be leveraged in the future to generate transplantable human tissues, building on current efforts to xenograft pig organs into humans with end-stage organ failure [160].

### 4.4. Biofunctionalization of the Microenvironment to Promote Organ Development

The extracellular matrix (ECM) is a dynamic and biologically active substrate which creates a biochemical and biophysical niche for living tissue [161]. This matrix consists of intricate networks of proteins, glycoproteins, lipids, nucleic acids, and polysaccharides, which provide structural support and serve as signaling nodes for cellular differentiation, migration, and other critical biological functions [162]. Cell adhesion receptors, such as integrins, specify cell–matrix interactions by recognizing distinct amino acid sequences within the ECM. RGD-binding integrins, for example, bind to the arginine, glycine, and aspartate residues present in fibronectin, vitronectin, fibrinogen, and laminin [163], while LDV-binding integrins recognize and attach to the leucine, aspartic acid, and valine sequence, which facilitates the interaction with fibronectin and vascular cell adhesion molecules [164]. The alpha domain of β1 integrins binds with high affinity to the GFOGER moiety present on collagen protein [165,166] and to laminin through a more complex binding interaction involving all five subunits of laminin proteins [166,167]. In living tissues, cells constantly remodel the matrix they inhabit by secreting matrix metalloproteinases (MMP), a family of zinc-dependent endopeptidases, that selectively degrade various components of the ECM, such as collagen and elastin [168]. Matrix remodeling helps establish the mechanical microenvironment and influences cell migration, proliferation, and differentiation [169].

By emulating these critical interactions within engineered tissue matrices, organoids can more faithfully emulate the complex architecture and function of their in vivo counterparts. For example, incorporating RGD motifs in neurite spheroid culture more accurately recapitulated in vivo neuritogenesis with organized out-spreading fibers [170], while GFOGER-containing hydrogels supported growth and passaging of enteroids and endometrial organoids [171]. Incorporating proteolytically degradable peptides into gels allows cells to remodel their microenvironment and increases the aggregate size, survival rate, and growth rate in ovarian follicle organoids [172]. Chaudhuri et al. generated a hydrogel that contained both RGD binding motifs and MMP-sensitive crosslinkers. They found that these modifications work in synergy to promote cellular spreading, proliferation, and differentiation of stem cells [44]. Novel cell-specific binding motifs can be identified using ultra-high-throughput chemical library synthesis, such as one bead one compound, and incorporated into culture substrates to target a specific cell type [173]. For example, Hao and others identified a new αvβ3 integrin ligand, LXW7, bound specifically to endothelial and endothelial progenitor cells [174], and leveraged this peptide to promote vascularized bone regeneration in rats and fetal sheep [175].

Although synthetic ECM designs with bioactive and bioresponsive components have significantly advanced modern organoid systems, our understanding of the matrix components that influence cellular behavior remains incomplete [176,177]; consequently, the use of native ECM has achieved considerable success in preclinical studies [176,177,178] and in clinical applications for soft tissue repair [177]. Decellularized ECM (dECM) can be prepared by perfusing native organs with detergents [178], enzymes [179], or physical methods such as ultrasonic cavitation to remove cellular material while preserving matrix components [180] (Figure 4A). dECM materials have been employed to enrich ex vivo culture of nearly organ system including the heart [181,182], brain [183], kidney [184], pancreas [185], skin [186], and in various cancers—including colon cancer [187] and cholangiocarcinoma [188]—to better understand the role of the matrix in physiological and pathological states [188]. A recent study by Sarig et al. showed that decellularized porcine cardiac ECM (pcECM) can initiate cell-specific behavior and organization in vitro. They found that human umbilical vein endothelial cells form monolayers on the pcECM surface, while mesenchymal stem cells penetrate deeper into the matrix. Co-cultures of these cells resulted in a synergistic effect, enhancing tissue integration and maturation, underscoring the significance of matrix composition in guiding cell behavior and tissue development [176]. The use of dECM has shown promise in a wide range of applications; however, inherent heterogeneity from donor variability [189] and differences in matrix composition across species leads to inconsistencies in preclinical and clinical applications [190,191]. Moreover, most human dECM materials are sourced from organs which are not suitable for transplantation and may not possess the full complement of ECM components necessary for optimal organ function, further complicating the use of dECM as a clinically applicable biomaterial [192].

### 4.5. Dynamic Microenvironments: Towards Spatiotemporal Control of the Biophysical and Biochemical Niches

Developing organs are exposed to rapidly evolving microenvironments, which are characterized by temporally controlled changes in size, shape, and mechanical forces such as compression and stretch [193]. For example, the volume of the human fetal liver increases 25,000 times its original size during embryonic days 26 to 56 [194], the time-dependent biaxial cyclic stretch induces lung mesenchyme development [82], and stiffness gradients modulate neural crest migration within a tightly regulated timeframe during in vivo neurulation [195]. Such biophysical processes are meticulously orchestrated in space and time, during a defined “competence window”, to direct cell fate decisions during organogenesis [196,197]. In neural tube organoids, applying an equibiaxial stretch during days 3–7 led to more effective patterning downstream of ECM production and planar cell polarity, compared to stretches applied outside of this competence window [196]. To better recapitulate the biophysical changes during organogenesis, researchers are utilizing “4D materials”, which possess mechanical properties that can be controlled through time via light [100]. Advances in photon laser technology have enabled researchers to achieve spatial resolutions in the sub-micron scale using two-photon laser scanning in photodegradable nitrobenzyl hydrogels [186]. Allyl sulfide hydrogels can be completely degraded in under 15 s, endowing a finer temporal resolution of gel modulation [197]. By tuning the timing, location, and intensity of the applied laser, one can precisely and reversibly tune the viscoelasticity, stiffness, shape, and size of the microenvironment through space and time [198,199].

Cell attachment can be spatiotemporally regulated within the niche using temperature-responsive materials like N-isopropylacrylamide [200] or by integrating light-sensitive RGD moieties into existing hydrogel systems [201] (Figure 4B). Tissue-level migration can be controlled using light-activated click chemistry [202] or acoustofluidics [203]. Ao et al. used sound waves to rotate and fuse multiple brain organoids, regulating neuronal projection and synapse formation through time [203]. Incorporating magnetic nanoparticles also provides spatiotemporal control of organoid growth and proliferation through the application of magnetic fields in the culture system [204]. In addition to the biophysical niche, the biochemical niche can also be tightly regulated through time to emulate the second-scale dynamics of morphogen gradients observed in vivo [205,206,207] by utilizing photo-reversible click chemistry [208] or microfluidic devices [209]. Signaling events can also be manipulated through cantilever-mediated deformation of the nucleus in live cells [210] or through the incorporation of light-activated proteins in cells or the microenvironment [211,212]. Optogenetics—a technique for controlling genetically modified, light-sensitive cells with light—offers millisecond resolution to study time-dependent biophysical and biochemical signaling during human embryonic development [211,213].

## 5. The Future of Organoid Clinical Applications

The greatest unmet challenge in organoid clinical applications is the development of structures that can engraft and replicate the complexity and function of adult human tissues. Current models predominantly exhibit fetal-like structure and function [214] and suffer from poor engraftment efficiency and survival after in vivo transplantation [215]. Accordingly, early clinical applications of organoids may focus on treating fetal and neonatal diseases, such as congenital diaphragmatic hernia (CDH), which can be diagnosed in the first trimester of pregnancy [216]. The in utero environment has evolved to transform populations of stem cells into organized organ structures. Organoids, derived from patient-specific cells, could be transplanted prenatally—allowing the tissues to develop with the fetus in the correct biochemical [217] and biophysical [218] niche for tissue growth and maturation. The potential for organoid-mediated regenerative therapies in adults hinges on gaining a more profound understanding of the biophysical and biochemical signals that drive organ maturation. Recent progress in organoid engraftment experiments, however, grants cautious optimism that tissue generated ex vivo can be applied in regenerative medicine in the coming decades. For example, in a recent work by Ma and others, airway organoids that can engraft, proliferate, and function in vivo for over two years were generated [219]. There are currently 205 clinical trials mentioning the term organoid [220]; however, current clinical research is limited to disease modeling, generally cancer or intestinal disease [220]. Because cancer cells from tumor biopsies can be cultured and expanded in vitro, it is possible to create personalized cancer models. These cancer models can be leveraged to generate targeted chimeric antigen receptor (CAR)-T therapies that are primed using patient-specific tumor cells prior to in vivo delivery [221].

While the clinical application of organoids remains elusive, organoid systems have advanced rapidly in the recent decade. Pioneering bioengineering teams have developed brain organoids that can “see” [145] or play computer games [222], implantable microelectrode arrays that can stimulate specific clusters of neurons [223,224], and biosensors that can quantify changes in neuronal activity [141]. Computational biology is growing exponentially and is helping us to derive meaning out of data with thousands of dimensions [225]. Bioengineered solutions, such as optogenetics, will help elucidate the complex interplay between biophysical and biochemical signaling events within cells, between cells, and throughout the cell–microenvironment interface [226]. As laser resolution continues to improve to the sub-nanometer range, we will be able to activate subdomains within organelles using optogenetic tools to investigate subcellular processes [116]. New photolabile crosslinkers can respond to distinct wavelengths within the visible light spectrum, enabling multiple dimensions of spatiotemporal control, without impacting cell viability [227]. These advances may permit us to investigate the spatiotemporally controlled biophysical and biochemical events underlying organ development.

The field of biophysics has advanced rapidly, in part because of the technological improvements in physical detection, enabling us to quantify forces in the zeptonewtons range (10^−21^ N) [228] and revealing a world of subcellular forces which had previously been invisible. Emerging fields in physics can similarly be applied in the realm of biology, elucidating forces and energies which are not yet adequately understood, such as quantum mechanics. For example, quantum coherence may help explain ion channel selectivity [229], quantum tunneling may influence neuronal signaling [230,231], quantum entanglement may influence spontaneous DNA mutations [232], and quantum computations in microtubules may influence what we perceive as consciousness [233]. Quantum modeling is currently being leveraged to design pre-emptive vaccines against pathogens before they undergo mutation by predicting spontaneous DNA mutations based on the electronic structures of molecular interactions [234]. Beyond quantum biology, other active areas of physics research may be similarly applied to better understand stem cell fate decisions and tissue-level organogenesis, including stochastic thermodynamics [235], non-equilibrium statistical physics [236], and topological physics [237]. The integration of modern physics into biology offers a promising avenue for scientific discovery and will help achieve a more profound understanding of biological phenomena including developmental events, healthy tissue maintenance, and seemingly stochastic disease mechanisms.

## Figures and Tables

**Figure 1 bioengineering-11-00619-f001:**
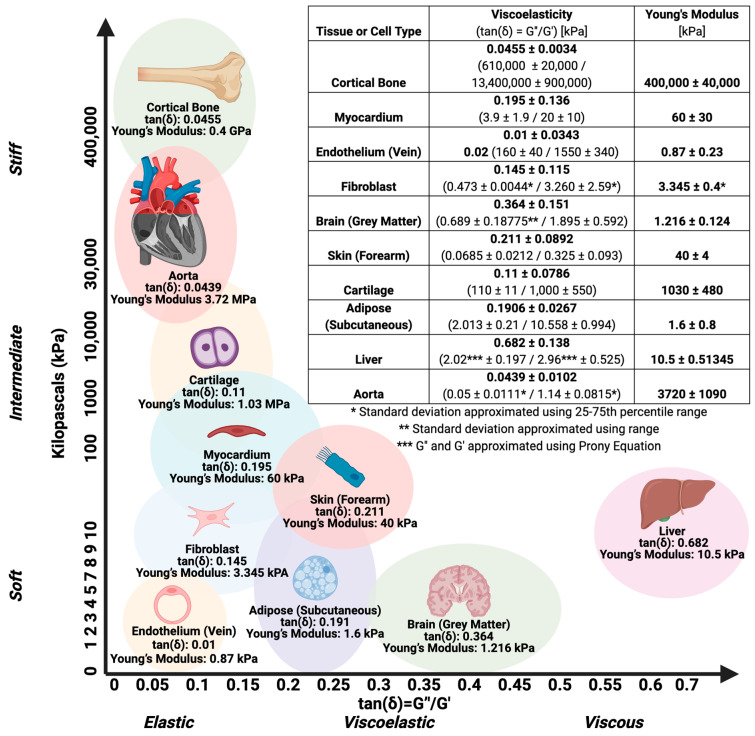
Viscoelasticity Versus Stiffness of Tissue. Viscoelasticity (tan(δ)) is calculated by dividing the Loss Modulus G″ (measure of viscosity) by the Storage Modulus G′ (measure of elasticity). More elastic materials will have a smaller tan(δ) (G″ < G′). Young’s modulus, a measurement of stiffness, is calculated by dividing the stress applied to a material (σ) by the responding strain (ε), or how much the material deforms. In this graph, we plot viscoelasticity against stiffness for native human tissues. tan(δ) was calculated by dividing G″/G′ at 1 Hz. Standard deviation of viscoelasticity and stiffness is represented by the area of the enclosing circle. Loss modulus, storage modulus, and Young’s modulus were acquired from the following references: Cortical Bone [47], Myocardium [48], Endothelium (Vein) [49,50], Fibroblast [51], Brain (Grey Matter) [52], Skin (Forearm) [53,54], Cartilage [55], Adipose (Subcutaneous) [56,57], Liver [58], and Aorta [59,60]. Notably, the instrumentation and techniques employed in the provided references varied across the studies. Consequently, these comparisons should be used with caution, as they provide a general overview rather than precise, direct comparisons across tissue types.

**Figure 3 bioengineering-11-00619-f003:**
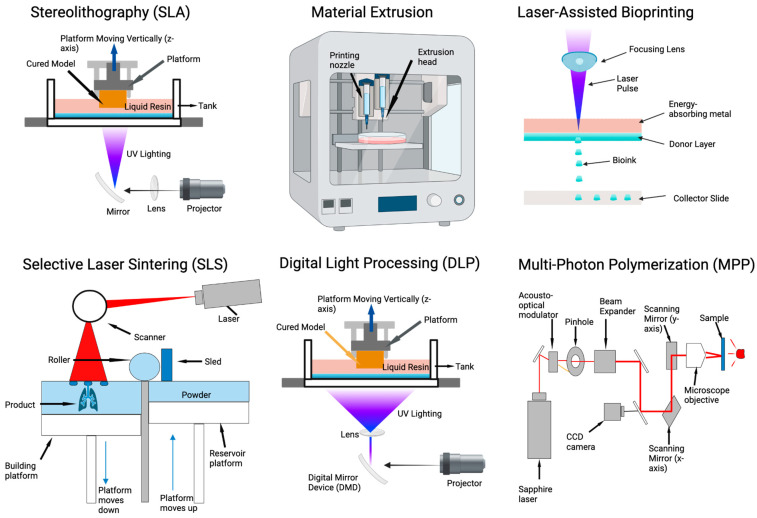
Three-dimensional Printing Methods for Tissue Modeling. Current approaches to tissue modeling. Multi-Photon Polymerization (MPP) currently offers the highest resolution, below 100 nm [116], and can be used to pattern cell-laden hydrogels with photocleavable subunits [117]. Current material extrusion (ME) approaches enable printing with multiple bioinks concurrently, allowing for the patterning of multiple cell types in 3D, albeit with a lower resolution than MPP [118].

**Figure 4 bioengineering-11-00619-f004:**
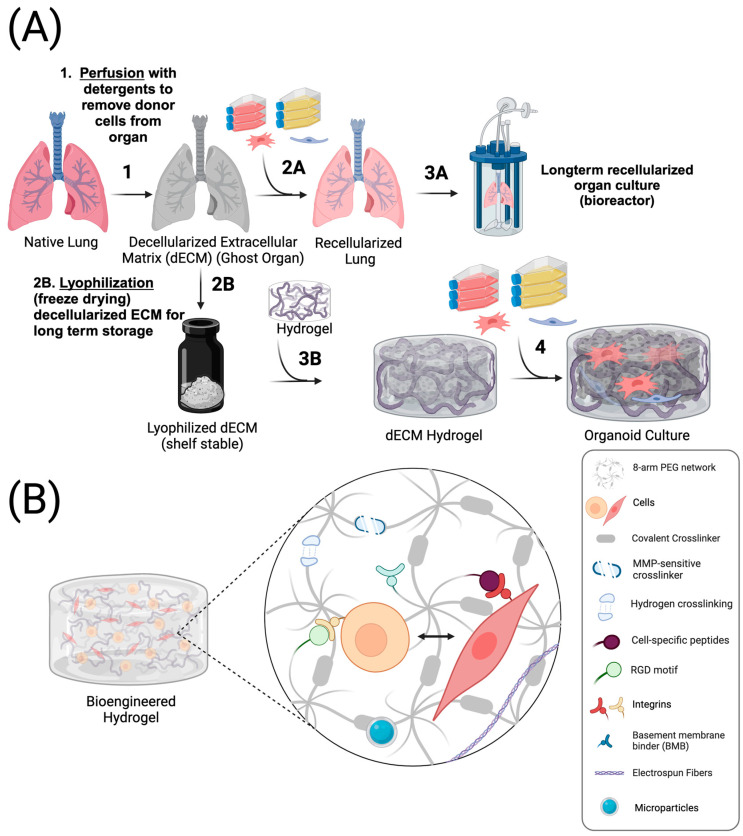
Engineered Hydrogels using Decellularized ECM or Synthetic Materials: (**A**) Fresh organs are perfused with detergents or other enzymes to lyse cell components while retaining the extracellular matrix (ECM) [178,179]. The resulting decellularized ECM (dECM) can be reseeded with patient-derived cells or lyophilized into a powder for long-term storage [182]. dECM can be later combined with hydrogels and retain some of the bioactive components [180,181,182,183]. (**B**) Hydrogels can also be designed with synthetic components to make engineered scaffolds. For example, peptides can be incorporated, which target integrin receptors, or matrix metalloproteinase (MMP)-sensitive crosslinkers can be added to facilitate cellular remodeling of the microenvironment [44].

## Data Availability

The only new data in this review is the derivation of viscoelasticity from G′ and G″ values of published papers that were cited. No new datasets were generated or analyzed during the current study.

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
