# Peer review of "The Role of Biophysical Factors in Organ Development: Insights from Current Organoid Models"

_bioengineering, 2024, doi:10.3390/bioengineering11060619_

Round 1

Reviewer 1 Report

Comments and Suggestions for Authors

This review is a very well written, succint and much needed highlighting the role of biophysical forces in organ development. The addition of organoid models that are not only important in studying organ development but also in cancer has been presented with enough examples and illustrations.

Author Response

The authors appreciate the kind words of encouragement. 

Reviewer 2 Report

Comments and Suggestions for Authors

The authors provide a review of biophysical regulation in organ development as evidenced by recent trends in research and bioengineering of organoids, microfluidics, 3D bioprinting, and quantum biology.

While there are several other reviews surveying individual aspects covered in this review the perspective and combination offered in this work is original, to the point, and provides understanding of current state-of-the-art, trends as well as future perspectives in bioengineering. Furthermore, the review is generally well written and easy to follow, however, there are still some major and minor revisions that need to be performed and adequately addressed prior to publication (detailed below). I therefore recommend that the authors be given the opportunity to revise the manuscript accordingly.

1.       Major comments:

a.    Quoted references do not always support the authors’ claims. For instance, Fig. 1 depicts the tangential and Young’s moduli for various tissues. Osteoblasts are described as having a Young’s modulus of 20 Gpa which is not logical for cells only, and the references quoted (5,6) are not at all dealing with this type of cell or measurement. The authors are requested to thoroughly double-check each and every reference quoted as well as the actual details and data that are quoted in the name of those references. Some other examples were identified in a random check and appear in subsequent clauses.

b.   Fig. 1 – while the attempt to graphically display the values of different tissues and cells in terms of elasticity/stiffness (Young’s Modulus) and viscoelasticity (tangential modulus), it should somehow be noted that:

                              i.    The values quoted are taken from different studies, and therefore deviation in methodologies, testing equipment and procedure, may yield only representative values and render this comparison somewhat inaccurate. In particular, the methods of calculating Young’s moduli (unitensile testers, compression testers, and AFM based testing) curtail varying sample preparation and orientation and hence also tend to yield different sets of values and value ranges. Also measuring tangential moduli of biomaterials and cells only in culture or within tissues from cadavers can be greatly affected by the sample source, handling, and testing method employed (oscillatory tested, AFM based, image analysis based etc.). Are all references quoted using the same methods? In the absence of a detailed materials & methods section (at least in the supplementary material I recommend adding it) it is very difficult to assess the comparability of these results, and in a sense, it may be like comparing apples and oranges...

                            ii.    Just for example, the aorta values point to ref. 43 as a source. This source, while providing a range of Young’s moduli (between ~3-9 MPa depending on orientation, were quoted in the figure as the maximal value only (~9Mpa) without mentioning other orientations of the aorta samples of lower values. The tangential moduli presented were never reported in ref. 43 to the best of this reviewer’s knowledge and so in the absence of alternative source, I cannot verify the calculation or the accuracy of this data.

                           iii.    The authors are requested to first of all double check the quoted values, consider presenting the average of several references and not just one (more details in the next clause), add some methodological details, and clearly add a few clarifications about such potential limitations of this graphical display approach to the text next to this figure.

                           iv.    If standard deviations/confidence intervals are presented, it may be better to display values based on the average of several studies and use the standard deviations as a way to compute a cloud of probabilities around these values. This may be more informative to the reader as opposed to the current icon-based appearance that is based on maximum values in each range.

c.    Another example of misquoting references relates to the mentioning of ref. 56 in terms of shear stress and its effect on kidney development and function. P4 lines 126-129: “Shear forces, transmitted by primary cilium, play a crucial role in determining the spatial arrangement of differentiating nephron segments during kidney development[56]. ” The word shear is mentioned only once in ref. 56 in the context of shear modulus measured for various hydrogels used to culture kidney organoids. There are no cilia on epithelial cells of the nephrons but rather villi which are completely different structures. Even in ref. 57, it refers to the effect of fluid velocity on villi (and not cilia) torque. Thus again the authors are paraphrasing information too broadly and not always accurately enough. The authors are, therefore requested to either provide stronger supportive evidence to their claims throughout the text and check again every reference and statement or change the text to describe exactly what the quoted references state.

d.   Fig. 2 is presented without providing any reference material to support the authors’ claims of the suggested and illustrated pathway, in particular with relation to the crowding/compression and spreading. The literature is abundant with descriptions of contradicting phenomena to that depicted in this figure. For instance in PMID 35012640, it is specifically stated that the “…spreading morphology of cells at low density, or exposure cells to stif matrices activated YAP” [i.e., dephosphorylates it to allow it to translocate into the nucleus]”, whereas compact morphology at high cell density, or shift cells from stif to soft matrices inhibited YAP activity” [i.e. via phosphorylation]. Similar statements are also stated in PMID 34109179. Collectively these statements seem to be  contradictory to the pathway presented by the authors. The authors are, hence, requested to provide scientific references to support these claims and if they can provide such references, describe the controversy between their suggested pathway and the one previously described by others. Specifically, with regards to the involvement of HIPPO vs. other activation pathways reported for YAP/TAZ.

e.    Similarly, the authors claim in Fig. 2 legend that “Cellular deformation results in molecular crowding…”etc., In particular, for the case of canonical WNT activation (i.e., via beta-catenin). While the authors did quote later on in text ref. 90, which supports their claim, on a more general observation, it contradicts other well documented phenomena. For instance, it is known that generally the HIPPO and the WNT pathway have an inhibitory relationship (e.g., PMID 34109179). This in general is the reason for which high-density cell cultures lead to inhibition of the WNT pathway via the HIPPO pathway. Please provide some more context as to this apparent controversy and or add appropriate references to discuss this accordingly and at least reflect the remaining knowledge gaps.

b)   Last couple of paragraphs seem to be only loosely related to the concept of the “future of clinical applications of organoids” as outlined by the section subtitle. In its current form, these paragraphs appear to be somewhat unrelated to this topic and go on to discuss hypothetical physics and its potential relationship with biology. Either omit these paragraphs altogether or change the subtitle/create a new section, and anyhow make a better linkage to the topic of quantum biology which context, in its current form, seems to be not sufficiently clear. Also, the abstract relatively has a long and non-proportional representation of the topic of ‘quantum biology’ which narrows down to two paragraphs at the end, but occupies over a fifth of the abstract.  

2.       Minor comments:

a.    Please revise overreaching/misleading statements. For example:
“…; however, we have only recently developed materials that can independently tune viscoelasticity without impacting stiffness, and vice versa[52].” – this sentence can be interpreted as if reference 52 quoted by the authors is a self-citation of their previous work, whereas in fact, this is work conducted in a completely different group.

b.   Similarly, P12 line 474 – P,13 line 477 “We have developed brain organoids that can “see”[139] or play computer games[209], implantable microelectrode arrays that can stimulate specific clusters of neurons[195,196] and biosensors that can quantify changes in neuronal activity[135].” – Please amend the misleading statement as all quoted references are not contributed by the authors themselves, but rather by other leading groups.

c.    In the context of the effect of remodeling of the extracellular matrix to produce neo tissue formation, please also quote PMID: 29500447.

d.   Please amend the term “transient receptor potential receptor [82]” into “transient receptor potential channels” (as also termed in the quoted ref. 82).

e.    P. 8 lines 297-298 – “vi) Multi-Photon Polymerization (MPP) is currently the most precise 3D printing technique capable of achieving less 100 nm resolution in droplet printing[125]” – please correct grammatically to “less than”.

Comments on the Quality of English Language

English level is excellent and the manuscript is well written in general (apart from a few minor comments here and there as specified above)

Reviewer 3 Report

Comments and Suggestions for Authors

The manuscript “The Role of Biophysical Factors in Organ Development: Insights from Current Organoid Models” by Wyle et al. showcases a well-structured and well thought of review article. Although organoids have been widely used since their inception in 2009, organoids, spheroids, microphysiological systems and other in vitro mammalian cell culture models  have become the need of the hour with the FDA Modernization Act 2.0 of 2022. The manuscript nicely covers the important aspects that researchers should consider while designing an organoid model. The importance of stiffness of the tissue, shape/size, mechanotransductive signalling pathways, approaches towards tissue modelling, biofunctionalization etc. I should not forget to mention that the figures are very engaging and self-explanatory, makes it easy on the eyes on the eyes of the reader to read more.

Just a minor concern in line 71 “….we have developed chemical cocktails….”, does the “we” here refer to the author and team?

Author Response

The authors are not responsible for the development of organoid chemical cocktails and have replaced "we" with "biologists" in line 71. 

Reviewer 4 Report

Comments and Suggestions for Authors

In this review, Wyle et al. provide an overview of the biophysical factors regulating organ development (such as stiffness, viscoelasticity, shear forces, mechanotransduction pathways and geometric constraints) and approaches to incorporate and study the effect of such biophysical parameters in organoid development in vitro. The review raises several interesting points. Nevertheless, the structure is erratic, and some sections are not clearly connected to the main scope. In addition to that, figures are not cited in the text, references in the text do not appear in the proper order and some abbreviations are not introduced (e.g. RGD, LDV, GFOGER), which is something that should be fixed.

In what follows, I provide some points I would like to discuss with the authors.

-              In the abstract, the authors mention that “transplanted tissue will be able to grow with the baby in the dynamic, fetal environment” (line 22). First, the word “baby” is not adequately used here. In addition, organoid transplantation in human embryos suggests generation of chimeric humans and this is a very serious endeavor with multiple ethical implications that requires debate with society stakeholders. I would suggest the authors the remove this and any reference to this potential experiment (which is brought back at the end of the introduction and in the discussion) from their review.

-              While fully agreeing with the authors that the knowledge provided by the frontiers of nowadays physics applied to biology can unveil how fundamental biological processes occur, I think that the focus of the authors on quantum physics appears off-topic. To make their discussion broader, the authors could mention other fields of physics — such as stochastic thermodynamics, non-statistical physics, or systems biology— which are as exciting than quantum physics, and have the same potential to provide understanding of biological processes.

-              The sentence spanning lines 16-17 in the abstract is not clear. What has yielded what?

-              In the abstract, the authors introduce the Young’s modulus. This concept does not come back (only in a figure caption, but not in the main text). Therefore, is it necessary? The authors should introduce what it is, what it is used for, how can we manipulate it, and what would be the impact.

-              The two sentences spanning lines 50-55 need to be supported by references.

-              In lines 88-92, the authors discuss the effect of increasing viscoelasticity in vitro. Later, in line 131 they mention how fluid shear stress has an effect in cyst development. It would add a lot of value to this review to discuss how to change viscoelastic properties and shear stress (as well as stiffness) in vitro. Part of this is briefly mentioned in lines 104-106.

-              Figure 1’s legend is difficult to read. I suggest the authors make a table with all the values for G’ and G’’, Y, tan(d), and references.

-              Figure 2 contains information about the YAP/TAZ mechano-transduction pathway but also molecular crowding. I would suggest the authors to split the contents of signaling and crowding in different panels. Crowding is only briefly discussed in the text, which also makes it harder to understand its relevance in the field of biophysical factors in organ development.

-              In the header of Section 4, are the authors talking about organs or organoids?

-              Section 4.1 requires an introduction to what the discussed technologies are doing. Are we seeding different cell types to print tissues with desired architectures? Also, overall, the structure of this section is confusing and repetitive; there is first an introduction to each technology and then bullet points repeating the same information.

-              As it is, it is not clear how Sections 4.2-4.3 are connected to the biophysical concepts introduced in the abstract and section 4.1. How do vascularized organoids relate to stiffness and shear forces?

-              It is a bit surprising that the role of Matrigel in tissue culture is not discussed next to dECM in the context of ECM stiffness. Matrigel, when used in gastruloids (an embryo model), demonstrated that while all gene programs were active, only with the suitable in vitro extracellular matrix (ECM) could these models develop more advanced features such as somites (PMID: 32076263, PMID: 36543322, PMID: 38405970). This was a very important finding in the field that should also be included in this review.

Comments on the Quality of English Language

As mentioned in the comments, the only issue if found in the abstract (lines 16-17). 

Reviewer 5 Report

Comments and Suggestions for Authors

The authors review the role of biophysical factors in organ development from the perspective of the design of organoid models. This is a very interesting subject that has multiple implications in current research, thus I sugest publication.

To improve the paper I  suggest the following:

1. Improve the introduction by indicating the limits of current review and citing other reviews on connected fields outside the limits of the current review. Also, reinforce the arguments to demonstrate that this review is new and needed.

2. The authors may wish to explore how wall shear stress induced by blood flow influences angiogenesis :  10.1039/d2lc00605g  https://doi.org/10.1073/pnas.1105316108 Wall shear stress has also a role in cell alignment, atherosclerosis and arteries remodeling. WSS is very important for a successful vascularisation.

Round 2

Reviewer 2 Report

Comments and Suggestions for Authors

The authors have addressed most of my comments, yet, several issues still need to be corrected prior to publication:

(1) A typo in one of the additions - p. 11 line 409, should be "various" (and not "varies")

(2) In the context of the effect of remodeling of the extracellular matrix to produce neo-tissue formation, please also quote PMID: 29500447.

(3) Regarding Fig. 1 - the second panel is actually a table, and maybe better presented as such, either in the main text, or in a supplementary file (i.e., not as a panel of a figure). 

(4) also regarding the table in Fig. 1 - if ranges could be described (i.e., either via min-max/standard deviation (SD) and/or confidence intervals) it would immensely benefit data interpretation from this table. This is particularly true also for the viscoelasticity (tan moduli) values. As the raw values used for calculations probably also contain some SD values in the original publications, these SD values could be dragged further or used on a relative scale to estimate the overall SD of the Tan moduli as well. Please add these SD values to the table accordingly.   

(5) once such ranges are added to the table, it may also inform the graphical representation (as suggested during my initial review comments) to also include indications of clouds of statistical ranges around the different tissues. 

(6) The bibliography list contains several different fonts, please make it more coherent. 

Round 3

Reviewer 2 Report

Comments and Suggestions for Authors

The authors have seriously responded to all my concerns. I have no further comments and recommend accepting the manuscript for publication in its present form. Good luck!